# Determinants of Learners' Self-Directed Learning and Online Learning Attitudes in Online Learning

Jing Li [1] and Chi-Hui Wu [2,*]

[1] College of Music and Film, Tianjin Normal University, Tianjin 300387, China; jenny934947047@gmail.com
[2] Department of Management and Information, National Open University, New Taipei 247031, Taiwan
[*] Correspondence: 940248@webmail.nou.edu.tw; Tel.: +86-158-229-77726

**Abstract:** The global COVID-19 pandemic has disrupted traditional learning methods, leading to a surge in online learning. It has been found that the low course completion and performance are associated with online learning. There has been increasing and urgent necessity to identify effective and decisive ways to address these challenges. Self-directed learning and online learning attitudes are key factors that influence learning behavior and outcomes, while the general traditional statistical method often does not perform well in identifying those categories. To fill the gap, this study applies the fuzzy Delphi method and the fuzzy decision-making trial and evaluation laboratory (DEMATEL) method to clarify and analyze the relationship of influence among indicators of self-directed learning and online learning attitudes, develop a cause–effect model, and ultimately identify an effective and decisive strategy for improving online learning. According to the cause–effect relationship among indictors, the computer/smartphone and internet confidence, computer/smartphone usage, and computer/smartphone preference are the three decisive strategical ways for online learning. To improve learners' attitudes towards online learning, teachers need to develop or improve students' computer/smartphone and internet confidence, computer/smartphone usage skills, and develop their self-directed learning abilities to inspire and increase their willingness and ability to participate effectively in online courses. Moreover, this study first applies the fuzzy DEMATEL method to assess, analyze and develop a causal model of self-directed learning and online learning attitudes for academics to further explore and confirm the complex interrelationships among the key learning behaviors of online learners.

**Keywords:** self-directed learning; online learning attitudes; fuzzy Delphi method; fuzzy DEMATEL

## 1. Introduction

Due to the COVID-19 epidemic, universities have adopted online teaching to allow students to study courses online. The internet is a common learning platform for learners and teachers to interact, communicate, and collaborate in a specific way [1,2], and the use of information technology (IT) in teaching has been implemented worldwide for decades. The purpose of developing online learning is to use IT to enhance the quality of teaching and learning, creating a high-quality learning environment, eliminating time and space constraints on learning, improving the management of teaching resources, and establishing the integration of IT with teaching and learning in various fields [3].

Since the 1990s, there has been an increase in the popularity of online education trends in mainland China [4–6]. Although online education has been promoted in mainland China for some time, it is still considered a supplementary tool to face-to-face teaching [7]. Due to lower information literacy in mainland China compared to Western countries [8] and lagging behind other countries in online education [9], online education in China has mostly failed when introduced by Western educational institutions [4–6], primarily due to the lack of core technology and perceived value [4,9].

In terms of technology, the Chinese government has been heavily promoting the development of online education infrastructure since 2018. Furthermore, due to the COVID-19 pandemic, educational institutions have strengthened their online teaching environments and required students to attend classes through online courses [9]. Although online education in mainland China has overcome technical issues, there is still room for improvement in learners' perceived learning [10], information literacy [8], and learning outcomes [11–13]. Therefore, this study aims to investigate the lack of perceived learning and information literacy as important factors among Chinese learners in online learning. It can help in the understanding of learners' self-directed learning abilities and attitudes toward online learning, serving as insights to enhance perceived learning and information literacy among learners.

During the COVID-19 pandemic, higher education institutions introduced the online teaching and learning mode. Students needed to devote more effort and energy to meet the requirements of the curriculums in the online learning environment [14]). Moreover, learners needed to spend more time on their studies, and their academic performance was not satisfactory [15].

Although learners' participation in online learning is a topic that has not been explored adequately [16], there is a large body of literature on the association of learning behaviors and online learning [17–19]. Learner's learning behavior is still a complex behavioral pattern and a complicated, multifaceted and uncertain concept [3]. Learners' online learning behavior includes self-directed learning [20–22], learning motivation [23–25], learning attitudes [26–28], learning engagement [29–31], and self-directed learning; online learning attitudes are important aspects of learning behavior [3], and the online learning behavior is the most crucial factor that affects learning outcomes [32], so when investigating an online learning environment for higher education, learners' self-directed learning on the learning behavior and online learning attitudes are the most important factors that are worth exploring.

Since learners' self-directed learning affects their motivation [33–37], learning attitudes [38–41], learning effectiveness [38,42–46], and that learning attitudes affect motivation [47–49], self-directed learning [44,50,51], learning engagement [52,53], learning satisfaction [54,55] and learning effectiveness [38,55–57]. Therefore, there is no consistent conclusion on the relationships between self-directed learning and online learning attitudes, and there are many different indicators of self-directed learning and online learning attitudes, and measuring self-directed learning and online learning attitudes requires one to consider multiple quantitative and qualitative criteria [3]. This study aims to clarify the relationships between learners' self-directed learning in online learning and the index of online learning attitudes that are so important to learners' learning willingness and capabilities in online learning.

In order to understand the complex relationships and determinants between learners' self-directed learning and online learning attitudes in online learning activities, this study uses the fuzzy Delphi method to survey scholars and experts on learning behavior to obtain indicators of self-directed learning and online learning attitudes and to establish a framework of self-directed learning and online learning attitudes. The study also analyzes the causal relationships among the dimensions and criteria of self-directed learning and online learning attitudes through the fuzzy DEMATEL method and identifies the determinants in order to provide educational institutions and schools with teaching strategies and curriculum designs for the integration of IT into online teaching. This study also constructs a causal model of self-directed learning and online learning attitudes for academics to further explore the complex interrelationships among the key learning behavior of learners who learn online, and to find out and enforce the crucial factors of learners' online learning behavior, upgrading learners' learning capabilities in online learning.

## 2. Literature Review

### 2.1. Self-Directed Learning and Online Learning Attitudes

Online learning has the capability of breaking down demographic boundaries and bringing together learners and teachers from various disciplinary backgrounds [58]. Consequently, online learning could be used at various branches of learning such as music teaching courses. Nowadays, online learning is ubiquitous and has transformed our way of thinking about teaching and learning [59]. Online learning has been applied to important leaning media and was used as a tool in higher education [58,60,61] because self-directed learning and online learning attitudes are important dimensions to learners' online learning behavior [3].

Self-directed learning is an effective learning method; it is flexible and not limited by the time and space, and learners can continuously enrich their professional knowledge, diagnosing their learning needs, find learning resources through their learning goals, and implement appropriate learning strategies to achieve learning outcomes [46]. Enhancing learners' self-directed learning can motivate learners to learn [46], and the higher the propensity for self-directed learning, the higher the satisfaction level of learners [46]; in addition, the higher the self-directed learning, the better the learning outcomes [45,46].

Researchers who study self-directed learning have different opinions, for instance, some scholars have adopted the readiness argument [20,62–65], and self-directed learning readiness refers to the attitudes, abilities, and attributes that one possesses when engaging in self-directed learning [63,65]. Scholars who have studied self-directed learning readiness have different views on its constitutive features. Fisher, King and Tague [62], Chen [63], and Kao, Yu, Kuo and Kuang [64] suggest that the constitutive features of self-directed learning readiness include self-management, desire for learning, and self-control. However, Chen [63] believes that the constitutive features of self-directed learning readiness include hope for the future, understanding of the self, active learning, self-confidence in learning things, and self-management. Deng [66], Chang and Chang [67], Shih, Chen and Huang [68] and Liang and Lai [69] suggest that the constitutive features of self-directed learning readiness are effective learning, enjoyment of learning, motivation of learning, active learning, independent learning, and creative learning.

Other scholars [43,70,71] have applied the ability argument, which suggests that self-directed learning ability affects online learning performances [43] and that self-directed learning ability is often seen as a valuable skill in school settings [43]. Tang, Zhu, Wen, Wang, Jin, and Chang [71] suggest that the constitutive features of the self-directed learning ability include self-management ability, information ability, and cooperative learning ability.

Some scholars (such as Knowles [72]) have introduced the learning contract theory, which is the process by which learners, with or without the assistance of others, can diagnose their own learning needs, set learning goals, identify learning resources, select and implement appropriate learning strategies, and evaluate learning outcomes [72]. Self-directed learning in the learning contract theory can be applied to the effective planning of teaching and learning [73], and emphasizes learner autonomy, the necessity of a two-way interaction between teachers and learners, and learner-centered teaching to develop learners' independent and autonomous learning skills [45].

This study integrates different perspectives on self-directed learning and classifies them into seven categories: self-learning efficacy, continuous learning, efficiency learning, independent learning, self-understanding, planned learning, and favorite learning.

Learning attitudes are determined by the interaction between learners and their surroundings during the learning process; therefore, the factors that influence learners' attitudes are complex [74]. Learning attitudes refer to learners' attitudes toward their interactions with the learning environment and depend on their abilities and experiences and their more persistent affirmative or negative behavioral tendencies or internal states toward learning things [75].

Online education is an important delivery method in various educational settings [76], and computer programs designed for education and the internet have fundamentally

changed university education [77], with learner attitudes affecting not only online teaching [78] but also learning satisfaction [54,55] and learning outcomes [38,56,57], while some other scholars argue that learning attitudes affect motivation in learning behavior [47–49,79] or self-directed learning [44,50]. Online learning is the use of computers and smartphones as media of transmission, providing a diverse teaching environment where different learners have different problems and attitudes when using computers for learning. Rainer and Miller [80] suggest that one of the most important factors in computer use should be the learner's attitude towards the computer, so building positive learning attitudes and computer skills can have a positive effect on the learner's learning outcomes. Hignite [81] argues that computer attitudes refer to learners' general perceptions of personal and social use of computers. This study has been conducted on learners who take online music lessons, so online learning attitudes are defined as the learners' willingness, interest, and emotional response to learning and interacting with computers and the internet, as well as their ability to use computers and the internet to equip and operate computers at speed.

This study focuses on the integration of IT into teaching and learning, where learners have to use computer devices and the internet to learn the content of music lessons; therefore, it refers to the computer attitude scale [26], the online teaching attitude scale [27], and the related online learning attitude studies [28]. The online learning attitudes are categorized into five components: computer/smartphone and internet confidence, internet usage, online learning interest, the usage and preference of computer/smartphone.

### 2.2. Fuzzy Delphi Method

When the understanding of a problem is not complete, the Delphi method is a suitable research tool [82]. The Delphi method solicits expert opinions through questionnaire surveys to obtain a consensus among experts [83,84], and presents the results using statistical methods. Reza and Vassilis [85] recommended a Delphi sample size of 10 to 15 participants, while Manoliadis, Tsolas and Nakou [86] concluded that a sample of fewer than 15 experts is optimal. Therefore, the number of experts should not be too high, and generally ranges from 15 to 20 [87].

The Delphi method is applicable in cases of insufficient data and uncertainty, where quantitative predictions are not possible [88]. It overcomes the weaknesses of qualitative research and leverages the objectivity and systematic nature of quantitative research [89]. The Delphi method is used in fields such as public policy and management [90], as it is a systematic process for expressing the opinions of expert groups [91].

However, the Delphi method requires repeated surveys to obtain consensus among experts, which increases implementation costs and survey time, resulting in a decrease in feedback. Additionally, the expression of opinions by different experts may lead to confusion. The fuzzy Delphi method can reduce the number of questionnaire surveys and fully express expert opinions, thereby saving time and reducing cost [84]. Moreover, the traditional Delphi method uses the arithmetic mean as the basis for evaluating criteria, which is susceptible to the influence of extreme values, leading to difficulty in reaching consensus. Klir and Yuan [92] proposed using the geometric mean of the generalization model as the consensus value of experts. That is, the geometric mean of the selection item is used as the consensus of experts. Therefore, the questionnaire is only administered once to obtain results, and then, the importance screening is performed. The fuzzy weights of each candidate factor are defuzzified into clear values using the simple gravity method, and the threshold value is set as the screening criterion. When using the fuzzy Delphi method to survey expert opinions, a minimum of 10 expert samples is required to obtain highly consistent views [84,93], usually ranging from 15 to 25 experts [84,94,95].

### 2.3. Fuzzy DEMATEL Method

The decision-making trial and evaluation laboratory (DEMATEL) is a tool used to construct a comprehensive solution for global and complex social issues, which involve

conflicting interests [96]. DEMATEL can display complex problems through structured causal relationships and has been successfully applied in various fields [97,98].

Tzeng, Chiang and Li [96] used a mixed multiple criteria decision analysis as a tool to construct an evaluation system for digital learning. In previous research, 20 criteria have been used as a basis for evaluation. When there are more than 20 criteria, factor analysis can be used to reduce them. Lin and Tzeng [99] suggested that threshold values should be established due to the causal diagram displayed in DEMATEL. The typical method is for experts to discuss and decide the threshold or for researchers to set it themselves. However, this could result in different causal diagrams among researchers. Afterwards, scholars began using an arithmetic mean as the threshold value [100].

This study aims to evaluate the relationship between self-directed learning and online learning attitudes by using the construct and criteria of these aspects. This involves the ambiguity and difficulty in quantifying learning behaviors in an online learning environment, leading to complexity and uncertainty in evaluating their factor relationships. According to Karwowski and Mital's [101] research, in many cases, it is unreasonable for experts to directly evaluate the possibility of an event occurring with a precise value. In fact, for a poorly defined event, experts can only use simple semantics, such as low, high, good, very good, etc., to evaluate the possibility or performance of the event. These semantics contain fuzziness, uncertainty, and multi-valuedness, posing a challenge to the evaluation of such problems. Conventional quantification methods cannot effectively apply to such fuzzy non-quantitative analysis. In fuzzy theory, experts can directly apply natural semantics to evaluate, and the semantics description can be converted into evaluative values of the relationship degree or occurrence possibility of different items through different membership function relationships. This allows evaluators to easily and fully express their subjective judgment values. Therefore, fuzzy theory is very suitable for analyzing and evaluating uncertain and fuzzy problems, and it has been widely used in multi-criteria management decision analysis. Thus, this study's fuzzy DEMATEL allows experts to directly apply natural semantics to evaluate the dimension and criteria relationships, which are then converted into evaluative values of the relationship degree using different triangular membership functions.

This study used the fuzzy Delphi method to screen the criteria and the fuzzy DEMATEL method to evaluate and analyze the causal relationships and strengths of the dimensions and criteria of self-directed learning and online learning attitudes. This identified the key learning behavioral factors that need to be prioritized for enhancement among learners.

In summary, the fuzzy DEMATEL method can be used to analyze and evaluate causality in complex problems, and reduce the ambiguity and uncertainty in expert evaluations through the use of fuzzy theory. This study applies this method to identify the factors that influence learners' online learning behaviors in online learning environments and provide suggestions for improving their learning behavior, which can help to enhance learning effectiveness and satisfaction.

## 3. Research Methodology and Design

In this study, the main factors of learners' self-directed learning and online learning attitudes, which are obtained from the literature review, are summarized, with a total of 12 criteria in two major dimensions. The main targets of the study are scholars and experts in western Taiwan, Beijing–Tianjin China, who study online learning (IT-assisted teaching).

Although the Pearson correlation coefficient test is able to explore and analyze the relationships between factors, it is not able to clarify what relationships exist among factors or define the complex relationships among factors [102]. Nevertheless, fuzzy DEMATEL is able to address these problems.

The study is conducted by using the fuzzy Delphi method first to select the criteria with higher relative importance and then the fuzzy DEMATEL method to explore the relationships among the dimensions and the criteria, constructing a matrix of the relationships among the dimensions and the criteria, drawing a cause–effect relationship diagram,

and analyzing the path of the cause–effect relationship. This study aims to explore the determinants of learner self-directed learning and the attitudes of learners in online learning.

This study uses the fuzzy Delphi method to identify relatively important criteria of self-directed learning and online leering attitudes. The fuzzy Delphi method [87,103] is a four-step process.

Step 1: Gather the views of the decision-making community.

It uses the linguistic variables of questionnaires to find out every expert's important assessment indexes for each criterion. As for the measuring criterion scales, Thomas [104] believes that three to seven scales are the most appropriate. This study uses a seven-point measuring scale for the linguistic scale to measure criterions affecting self-directed learning and online learning attitudes and to adopt the geometric mean for integrating every expert's opinion [105].

Step 2: Create a triangular fuzzy number.

To calculate experts' important triangular fuzzy numbers, this study applies the geometric mean of the general mode for an average mean that Klir and Yuan [92] developed, namely the approach that fuzzy DEMATEL uses to count the group's consensus decision making.

Step 3: Defuzzification.

Because fuzzy numbers are not clear values and are not compared directly, we need defuzzification for fuzzy numbers. The purpose of the process of defuzzification is to find out the best non-fuzzy performance value, BNP. This study calculates BNP in accordance with the graded mean integration that Chen and Hsieh [106] developed.

Step 4: Selection of evaluation criteria.

Threshold values and consistently statistical judgement standards of the experts' opinions need to be set up for selection and assessment criteria [107]. Using threshold values helps in the selection of much more appropriate criteria. In general, 60% to 80% of the maximum value is adopted [87]. A total of 70% of the maximum value is the threshold value to be applied in this study [3].

The retention dimensions and criteria questionnaires were distributed to 20 academics and practical experts with more than ten years of experience in studying online music learning programs at universities, using knowledge and experience to determine whether to retain a criterion. The threshold used in this study is 70% [108], meaning that the criterion will be kept if more than 70% of academics and experts agree to keep it. The two dimensions and 12 criteria identified in this study have all been kept because more than 70% of experts and academics agreed to keep them, as shown in Tables 1 and 2.

**Table 1.** Statistics of scholars and experts in online learning.

| Years | Attribute | Working Place | City |
|---|---|---|---|
| 15 | Digital distant learning | Beijing Normal University | Beijing, China |
| 25 | Educational evaluation | Beijing Education University | Beijing, China |
| 24 | Music blended teaching | Central Conservatory of Music | Beijing, China |
| 18 | Finance blended teaching and learning | Tianjin University of Finance and Economics | Tianjin, China |
| 26 | Teaching materials and methods of technical and vocational education | Tianjin Art Vocational College | Tianjin, China |
| 22 | Inquiry and practice teaching, learning, and curriculum design | Tianjin Normal University | Tianjin, China |
| 12 | Teaching design and research on creative thinking ability | Beijing Jiaotong University | Beijing, China |
| 28 | Digital education and teaching management environment | Beijing Tsinghua University | Beijing, China |
| 16 | Digital teaching and learning research | Tianjin University | Tianjin, China |

**Table 1.** *Cont.*

| Years | Attribute | Working Place | City |
|---|---|---|---|
| 19 | IT-assised music teaching and learning | Tianjin Normal University | Tianjin, China |
| 23 | Digital learning instructional design | National University of Tainan | Tainan, Taiwan |
| 17 | Online education curriculum research | National Cheng Kung University | Tainan, Taiwan |
| 26 | Art blended teaching and learning | Tunghai University | Taichung, Taiwan |
| 27 | Integration of IT into teaching | National Taichung University of Education | Taichung, Taiwan |
| 22 | Online IT teaching and learning research | Da-Yeh University | Changhua, Taiwan |
| 16 | Educational psychology research | National Changhua University of Education | Changhua, Taiwan |
| 18 | Design, production, and evaluation of educational technology integrated into teaching | National Tsing Hua University | Hsinchu, Taiwan |
| 15 | Marketing management blended teaching | Chinese Culture University | Taipei, Taiwan |
| 23 | Online art teaching research | Taipei National University of the Arts | Taipei, Taiwan |
| 21 | IT-assised technology teaching and learning | National Taipei University of Technology | Taipei, Taiwan |

**Table 2.** Fuzzy Delphi method questionnaire item statistics.

| No. | Self-Directed Learning (S)/Online Learning Attitudes (O) | Thresh Hold (Fuzzy Performance Values) | Retain/Delete |
|---|---|---|---|
| 1 | Self-learning efficacy (S1) | 0.870 | Retain |
| 2 | Continuous learning (S2) | 0.889 | Retain |
| 3 | Efficiency learning (S3) | 0.726 | Retain |
| 4 | Independent learning (S4) | 0.744 | Retain |
| 5 | Self-understanding (S5) | 0.844 | Retain |
| 6 | Planned learning (S6) | 0.898 | Retain |
| 7 | Favorite learning (S7) | 0.825 | Retain |
| 8 | Computer/smartphone and internet confidence (O1) | 0.836 | Retain |
| 9 | Internet useage (O2) | 0.879 | Retain |
| 10 | Online learning interest (O3) | 0.870 | Retain |
| 11 | Computer/smartphone useage (O4) | 0.853 | Retain |
| 12 | Computer/smartphone preference (O5) | 0.799 | Retain |

The fuzzy DEMATEL is a method that combines fuzzy semantic variables and the DEMATEL method. The formula and calculation steps [100,109–111] have seven steps, which are as follows:

Step 1: Define the evaluation criteria and design a fuzzy semantic scale.

The evaluation criteria are shown as $C = \{ C_i | i = 1, 2, \ldots, n \}$, and the fuzzy linguistic scale, taken from Li, Wu, Chen, Huang and Lin [112], is divided into Very High Effect (VH), High Effect (H), Low Effect (L), Very Low Effect (VL), and No Effect (No).

Step 2: Create a direct association matrix.

The initial fuzzy direct association matrix $\widetilde{Z}$, below, is obtained by having the participants (experts) carry out comparisons between pairs of criteria.

$$\widetilde{Z} = \begin{matrix} C_1 \\ C_2 \\ \vdots \\ C_n \end{matrix} \begin{bmatrix} 0 & \widetilde{Z}_{12} & \cdots & \widetilde{Z}_{1n} \\ \widetilde{Z}_{21} & 0 & \cdots & \widetilde{Z}_{2n} \\ \vdots & \vdots & \ddots & \vdots \\ \widetilde{Z}_{n1} & \widetilde{Z}_{n2} & \cdots & 0 \end{bmatrix} \quad (1)$$

$\widetilde{Z}_{ij} = (l_{ij}, m_{ij}, r_{ij})$ are triangular fuzzy numbers, and $Z_{ii}, i = 1, 2, \ldots, n$, on the diagonal is (0, 0, 0).

Step 3: Build and analyze the structural model.

The linear scale is changed to a normalization formula so that the criteria scale can be transformed into comparable scales:

$$\widetilde{a}_{ij} = \sum_{j=1}^{n} \widetilde{Z}_{ij} = \left( \sum_{j=1}^{n} l_{ij}, \sum_{j=1}^{n} m_{ij}, \sum_{j=1}^{n} r_{ij} \right) \text{ and } r = \max_{1 \leq i \leq n} \left( \sum_{j=1}^{n} r_{ij} \right) \tag{2}$$

based on *X*, the normalized direct association fuzzy matrix is established as $\widetilde{X} = r^{-1} \otimes \widetilde{Z}$, so that

$$\widetilde{X} = \begin{bmatrix} \widetilde{X}_{11} & \widetilde{X}_{12} & \cdots & \widetilde{X}_{1n} \\ \widetilde{X}_{21} & \widetilde{X}_{22} & \cdots & \widetilde{X}_{2n} \\ \vdots & \vdots & \ddots & \vdots \\ \widetilde{X}_{m1} & \widetilde{X}_{m2} & \cdots & \widetilde{X}_{mn} \end{bmatrix} \text{ and } \widetilde{X}_{ij} = \frac{\widetilde{Z}_{ij}}{r} = \left( \frac{l_{ij}}{r}, \frac{m_{ij}}{r}, \frac{r_{ij}}{r} \right) \tag{3}$$

Step 4: Total association matrix.

After establishing the normalized direct association matrix $\widetilde{X}$, the fuzzy total association matrix $\widetilde{T}$ can be established using the following equations.

$$\begin{aligned} \widetilde{T} &= \widetilde{X} + \widetilde{X}^2 + \cdots + \widetilde{X}^k \\ &= \widetilde{X}\left( I + \widetilde{X} + \widetilde{X}^2 + \cdots + \widetilde{X}^{k-1} \right) \\ &= \widetilde{X}\left( I + \widetilde{X} + \widetilde{X}^2 + \cdots + \widetilde{X}^{k-1} \right)\left( I - \widetilde{X} \right)\left( I - \widetilde{X} \right)^{-1} \\ &= \widetilde{X}\left( I - \widetilde{X} \right)^{-1}, \text{when } \lim_{k \to \infty} \widetilde{X}^k = [0]_{nxn} \end{aligned}$$

$$\widetilde{T} = \begin{bmatrix} \widetilde{t}_{11} & \widetilde{t}_{12} & \cdots & \widetilde{t}_{1n} \\ \widetilde{t}_{21} & \widetilde{t}_{22} & \cdots & \widetilde{t}_{2n} \\ \vdots & \vdots & \ddots & \vdots \\ \widetilde{t}_{m1} & \widetilde{t}_{m2} & \cdots & \widetilde{t}_{mn} \end{bmatrix} \text{ and } \widetilde{t}_{ij} = \left( l_{ij}'', m_{ij}'', r_{ij}'' \right) \tag{4}$$

$$\left[ l_{ij}'' \right] = X_l \times (I - X_l)^{-1}$$
$$\left[ m_{ij}'' \right] = X_m \times (I - X_m)^{-1}$$
$$\left[ r_{ij}'' \right] = X_r \times (I - X_r)^{-1}$$

Step 5: Conduct defuzzification.

From Equation (5), the fuzzy numbers can be defuzzified to obtain the total association matrix *T*.

$$\mathrm{d}F_{ij} = \frac{(r_{ij} - l_{ij}) + (m_{ij} - l_{ij})}{3} + l_{ij} \tag{5}$$

Step 6: Centrality and causality.

The row and column values are acquired using Equation (6) and are defined as *d* and *r*.

$$T = \left[ t_{ij} \right], i, j \in \{1, 2, \ldots, n\}$$

$$d = (d_i)_{n \times 1} = \left[ \sum_{j=1}^{n} t_{ij} \right]_{n \times 1} ; r = (r_j)'_{1 \times n} = \left[ \sum_{i=1}^{n} t_{ij} \right]'_{1 \times n} \tag{6}$$

Step 7: Result analysis.

The purpose of the fuzzy DEMATEL analysis is to assess the cause–effect relationships among factors and to establish a structural model. According to the definition of the causal diagram in the fuzzy DEMATEL analysis, the causal diagram among factors can be

acquired by mapping the dataset of the value $(d + r)$ and $(d - r)$, where the horizontal axis $(d + r)$ is made by adding $d$ to $r$, and the vertical axis $(d - r)$ is made by subtracting d from $r$.

After calculating $(d + r)$ and $(d - r)$, a diagram of the correlations among the criteria can be drawn. $(d + r)$ represents the effects among the criteria, with a higher value signifying a greater effect. $(d - r)$ represents the causal relations among the criteria, with a higher value indicating that the criteria are the causes of other criteria, and a lower one indicating that they are the results of other criteria.

The influence–relation map, which indicates the cause–effect relationship among factors, can be established based on the total relation matrix *T*. To avoid over-complicated causality when drawing the influence–relation map, the decisionmaker group should set up a threshold value to filter out some negligible relationships. This enables the decision maker to choose only the relationships greater than the threshold value and to map the cause–effect relationship accordingly.

## 4. Analysis and Discussion of the Findings

In this stage, 20 scholars and practical experts with more than ten years of experience studying online music learning programs at universities were invited to take the survey. The questionnaires were distributed to these researchers and practitioners for completion. After conducting the survey for three months, there were 20 valid questionnaires, including 10 from researchers and 10 from practitioners. The results of the various dimensions and criteria were analyzed.

### 4.1. Results of the Analysis of Each Dimension

In this stage, 20 scholars and practitioners with more than ten years of experience studying online music learning programs at universities were invited to take the survey. The questionnaires were distributed to these researchers and practitioners for completion. After conducting the survey for three months, there were 20 valid questionnaires, including 10 from researchers and 10 from practitioners. The results of the various dimensions and criteria were analyzed. The evaluative dimensions were self-directed learning (S) and online learning attitudes (O). First, the evaluative dimensions were defined, which resulted in the design of the fuzzy semantic scales, the establishment of the direct association matrix, the creation and analysis of the structural model, the creation of the total association matrix, and defuzzification. The formulae, calculations and the defuzzification matrix of each dimension are shown in Table 3. The column and row values of each dimension after the calculation of the centrality and causality are shown in Table 4. Then, after obtaining the values of $d + r$ (centrality) and $d - r$ (causality), the cause–effect diagram can be plotted for each value, as shown in Figure 1. The value of $d + r$ (centrality) represents the strength of the influence between the dimensions (the higher the value, the stronger the influence). When the value of $d - r$ is positive and if the value is higher than the threshold, it represents the "cause" of the influence of other dimensions, and when $d - r$ is negative and if the value is lower than the threshold, it represents the "effect" of the influence of other dimensions.

**Table 3.** Matrix of defuzzied total correlations of the dimensions.

| Dimension | Self-Directed Learning (S) | Online Learning Attitudes (O) |
|---|---|---|
| Self-directed Learning (S) | 6.941 | 7.558 * |
| Online Learning Attitudes (O) | 7.094 | 6.941 |

Note: * indicates value above the threshold value of 7.133.

**Table 4.** Collation of column and row values of dimension.

| Dimension | $d$ (Column Values) | $r$ (Row Values) | $d + r$ (Centrality) | $d - r$ (Causality) | $d/r$ (Influence Ratio) | Causal Relationship |
|---|---|---|---|---|---|---|
| S | 14.499 | 14.035 | 28.534 | 0.464 | 1.033 | Affects another dimension |
| O | 14.035 | 14.499 | 28.534 | −0.464 | 0.968 | Independence dimension |
| Average | | | 28.534 | 0 | | |

Note: Self-directed learning (S), online learning attitudes (O).

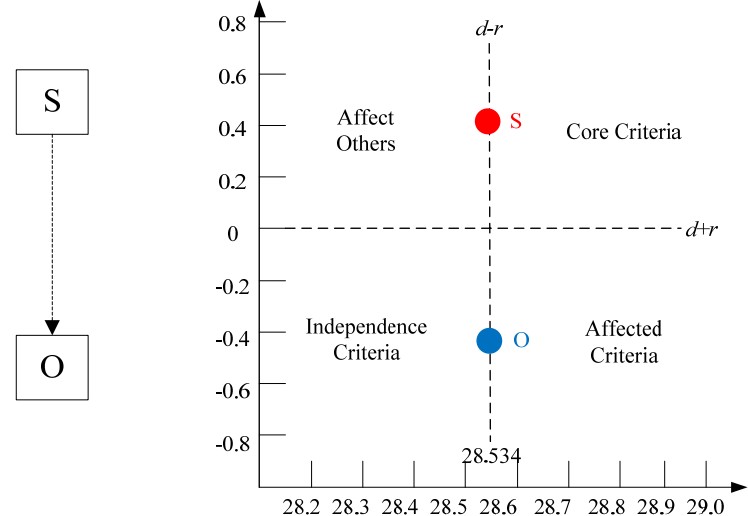

**Figure 1.** Causality correlation diagram of each dimension. Note: self-directed learning (S), online learning attitudes (O).

In the causality ($d - r$) section, according to the value of $d - r$ (causality), the dimensions of self-directed learning and online learning attitudes are classified into cause and effect clusters. The dimensions with positive $d - r$ (causality) values are classified as cause groups. The positive value of the self-directed learning (S) dimension directly affects another dimension. Therefore, schools, educational institutions, and teachers should consider this dimension as the main dimension in developing learners' learning behavior in online learning programs.

The main purpose of learners' learning behavior is to enforce the dimension in cause groups, namely self-directed learning, so as to improve self-directed learning. Hence, self-directed learning (S) is the strongest affecting dimension and should be listed as the main dimension, which could strengthen a learner's learning behavior, while the online learning dimension, which has negative $d - r$ (causality) values, is categorized as an effect cluster (O). This means that it is affected by others, and the extent to which this dimension was affected is greater than its own influence, so schools, educational institutions, and teachers can, therefore, consider the online learning attitude dimension as a problem to be solved in the long-term development of learners' learning behavior. The highest positive value of $d - r$ is self-directed learning (S), which represents the "cause" of the largest influence on the other dimensions, while online learning attitudes (O) are the "effect" of the largest influence from other dimensions. The higher the value of self-directed learning (S), the stronger the online learning attitudes (O). Hence, the self-directed learning dimension is the foundation of the learner's learning behavior.

In terms of overall consideration, if learners want to improve their learning behavior at an online learning course, they should choose the most influential dimension, namely "self-directed learning (S)", which directly affects the dimension ("online learning attitudes") (O).

The arithmetic mean of centrality ($d + r$) is 28.534, and it is set as the threshold value. The self-directed learning dimension is located in Quadrants 1 and 2, while online learning attitudes are in Quadrants 3 and 4. As shown in Figure 1, the dimension of self-directed learning in the first and second quadrants occupies a relatively important position in the other quadrant and affects the dimension in the third and fourth quadrants, where the dimension of online learning attitudes is. As this dimension does not affect the self-directed learning of the learner, it is listed as the least important dimension, and it could be strengthened through the dimensions in the first and second quadrants.

In Figure 1, we can observe that self-directed learning (S) affects online learning attitudes (O), and it is clear that the direction of the arrow of self-directed learning (S) points directly towards the online learning attitudes (O). Hence, learners should cultivate their self-directed learning to enforce their learning attitudes in order to perfect their online learning course behavior.

The results of this study are consistent with previous research conducted by Ames and Archer [38], Faisal and Eng [39], Zhang et al. [40], and Chen et al. [41], which indicate that the self-directed learning dimension has an impact on the consistency of online learning attitudes. Therefore, self-directed learning is a crucial dimension that determines learners' learning behavior in online learning environments and affects their online learning attitudes. Self-directed learning refers to learners' ability and skills to set learning goals and utilize learning resources, which has a significant impact on learners' learning behavior and learning outcomes. Autonomous and effective learning is essential for learners' learning behavior and outcomes.

### 4.2. Results of the Analysis of the Criteria

The assessment criteria are self-learning efficacy (S1), continuous learning (S2), efficiency learning (S3), independent learning (S4), self-understanding (S5), planned learning (S6), favorite learning (S7), computer/smartphone and internet confidence (O1), internet usage (O2), online learning interest (O3), computer/smartphone usage (O4), and computer/smartphone preference (O5). There is a total of 12 criteria. After defining the criteria and designing the fuzzy semantic scale, establishing a direct association matrix, building and analyzing the structural model, the total association matrix, and defuzzification, the defuzzified total association matrix among the criteria is shown in Table 5. Once $d + r$ (centrality) and $d - r$ (causality) have been obtained, the cause–effect relationship diagram can be plotted against each value, as shown in Figure 2.

**Table 5.** Collation of column and row values of criteria.

| Criteria | $d$ | $r$ | $d + r$ | $d - r$ | $d/r$ | Causal Relationship |
|---|---|---|---|---|---|---|
| S1 | 4.876 | 4.418 | 9.293 | 0.458 | 1.104 | Cause (core) criteria |
| S2 | 4.403 | 4.428 | 8.831 | −0.024 | 0.994 | Independence criteria |
| S3 | 4.329 | 4.491 | 8.820 | −0.162 | 0.964 | Independence criteria |
| S4 | 3.757 | 4.462 | 8.219 | −0.705 | 0.842 | Independence criteria |
| S5 | 4.033 | 4.464 | 8.497 | −0.430 | 0.903 | Independence criteria |
| S6 | 4.042 | 4.623 | 8.665 | −0.581 | 0.874 | Independence criteria |
| S7 | 4.252 | 4.653 | 8.905 | −0.401 | 0.914 | Independence criteria |
| O1 | 4.961 | 4.497 | 9.458 | 0.464 | 1.103 | Cause (core) criteria |
| O2 | 4.075 | 4.530 | 8.605 | −0.456 | 0.899 | Independence criteria |
| O3 | 4.966 | 4.395 | 9.361 | 0.570 | 1.130 | Cause (core) criteria |
| O4 | 4.916 | 4.403 | 9.319 | 0.512 | 1.117 | Cause (core) criteria |
| O5 | 5.140 | 4.385 | 9.525 | 0.754 | 1.172 | Core criteria |
| Average | | | 8.958 | 0 | | |

Note: self-learning efficacy (S1), continuous learning (S2), efficiency learning (S3), independent learning (S4), self-understanding (S5), planned learning (S6), favorite learning (S7), computer/smartphone and internet confidence (O1), internet usage (O2), online learning interest (O3), computer/smartphone usage (O4), computer/smartphone preference (O5).

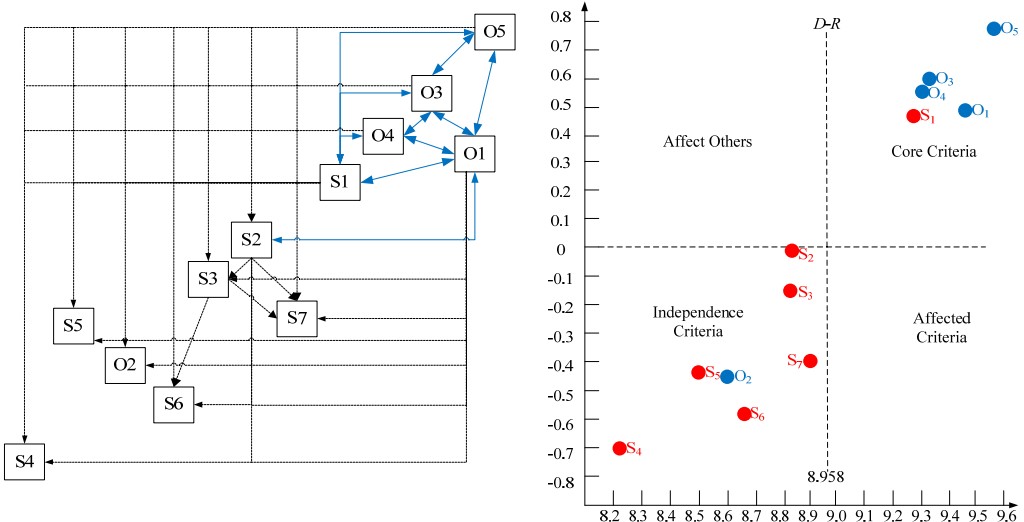

**Figure 2.** Cause–effect relationship diagram. Note: self-learning efficacy (S1), continuous learning (S2), efficiency learning (S3), independent learning (S4), self-understanding (S5), planned g learning (S6), favorite learning (S7), computer/smartphone and internet confidence (O1), internet usage (O2), online learning interest (O3), computer/smartphone usage (O4), computer/smart phone preferences (O5).

In terms of centrality ($d + r$), these three criteria (computer/smartphone and internet confidence (O1), online learning interest (O3), and computer/smartphone preference (O5)) are the most important. In terms of the causality ($d - r$), the value of these criteria (self-learning efficacy (S1), computer/smartphone and internet confidence (O1), online learning interest (O3), computer/smartphone usage (O4), and computer/smartphone preference (O5)) are positive, which means that these are the cause criteria. Among them, the strongest are online learning interest (O3), computer/smartphone usage (O4), and computer/smartphone preference (O5). Conversely, the values of these seven criteria—continuous learning (S2), efficiency learning (S3), independent learning (S4), self-understanding (S5), planned learning (S6), favorite learning (S7), and internet usage (O2)—are negative, which means that these criteria are effect criteria. Among these criteria, independent learning (S4), planned learning (S6), and internet usage (O2) have the highest negative values.

According to the causal relationships obtained from the combined centrality and causality analyses, computer/smartphone preference (O5) have the strongest influence, while the most influential criterion is independent learning (S4). Among the criteria of self-directed learning and online learning attitudes, online learning interest (O3), computer/smartphone usage (O4), and computer/smartphone preference (O5) are the most influential criteria and are the main criteria for improving learner's self-directed learning and online learning attitudes.

In the causality ($d - r$) section, the 12 criteria of self-directed learning and online learning attitudes can be grouped into cause–effect clusters based on the $d - r$ (causality) values. Criteria with positive $d - r$ (causality) values are categorized as cause clusters, with a total of five criteria categorized. Positive criteria have a direct impact on other criteria. Therefore, scholars should consider these criteria as important targets for enhancing self-directed learning and online learning attitudes and strengthening the ability of the criteria of the cause group to enhance other criteria of self-directed learning and online learning attitudes. The most influential criteria are "online learning interest (O3), computer/smartphone usage (O4), and computer/smartphone preference (O5)". These three criteria are the most influential criteria and should be treated as the most important criteria for self-directed learning and online learning attitudes and the most influential "cause" of the other criteria. The higher the proportion of online learning interest, computer/smartphone usage, and com-

puter/smartphone preference, the stronger the influence of other criteria on self-directed learning and online learning attitudes. Therefore, the learners' online learning interest, computer/smartphone usage, and computer/smartphone preference are the basis for self-directed learning and online learning attitudes. The negative value of $d − r$ (causality) is classified as the effect cluster. A total of seven criteria are categorized as "effect clusters," representing the extent to which they are influenced by other criteria. The extent affected by these seven criteria is greater than their own influence; therefore, schools, educational institutions, and teachers can consider these seven criteria as the long-term development of learners' self-directed learning and online learning attitudes to be addressed in online learning programs.

The arithmetic mean of centrality ($d + r$) is 8.958, and its value is set as the threshold value. The criteria of self-directed learning and online learning attitudes fall in the first and third quadrants, as shown in Table 4. These five criteria are self-learning efficacy (S1), computer/smartphone and internet confidence (O1), online learning interest (O3), computer/smartphone usage (O4), and computer/smartphone preference (O5), which means that they have a high degree of centrality and causality. The criteria of self-directed learning and online learning attitudes in this quadrant are relatively important compared to the criteria in other quadrants. They influence the criteria in Quadrants 2 and 4, and are, therefore, important criteria for self-directed learning and online learning attitudes. The criteria for self-directed learning and online learning attitudes in Quadrant 3 are continuous learning (S2), efficiency learning (S3), independent learning (S4), self-understanding (S5), planned learning (S6), favorite learning (S7), and internet usage (O2). These are the criteria with low centrality and low causality, so they are classified as the least important criteria for learners' self-directed learning and online learning attitudes. These seven criteria in Quadrant 3 can be improved through the criteria in Quadrant 1 so as to promote learners' self-directed learning and online learning attitudes. This means that improving the criteria of Quadrant 1 will improve the criteria in Quadrant 3. As the dimension of self-directed learning affects the criteria of online learning attitudes, there is a complex entanglement between self-directed learning and the criteria of online learning attitudes. Learners can strengthen their self-directed learning and online learning attitudes by addressing the criteria of self-learning efficacy (S1), computer/smartphone and internet confidence (O1), online learning interest (O3), computer/smartphone usage (O4), and computer/smartphone preference (O5), which are listed as the first priorities for enhancing self-directed learning and online learning attitudes.

It can be observed that online learning interest, computer or smartphone use, and computer or smartphone preferences are the determinants of self-directed learning and online learning attitudes. They influence the other seven criteria of self-directed learning and online learning attitudes. Alternatively, self-directed learning will affect attitudes toward online learning. The self-learning efficacy criterion is the main criterion for self-directed learning. The self-learning efficacy criterion is the key criterion to self-directed learning, and it is related to computer/smartphone and internet confidence (O1), online learning interest (O3), computer/smartphone usage (O4), and computer/smartphone preference (O5). Therefore, self-learning efficacy should also be considered a key factor in enhancing self-directed learning and online learning attitudes, as it has a reciprocal effect on other criteria for self-directed learning.

The results of this study show that there is a complex relationship between the criteria of self-directed learning and online learning attitudes, meaning that the criteria can mutually influence each other. Previous studies have also pointed out that self-directed learning criteria affect online learning attitude criteria, while online learning attitude criteria partially affect self-directed learning criteria. Since self-learning efficacy, computer/smartphone and internet confidence, online learning interest, computer/smartphone usage, and computer/smartphone preferences are decisive criteria for self-directed learning and online learning attitudes, online learning attitudes refer to learners' confidence in and willingness to use information terminal devices and the internet for online learning, as well as their

level of interest in information terminal devices. Therefore, learners need to cultivate online learning attitudes to adapt to the online learning environment [15,113]. In addition, the application of information technology into online learning requires learners to have self-directed learning abilities [114,115] in order to enhance academic performance [115].

Based on the analysis of the dimensions and criteria of self-directed learning and online learning attitudes, higher education institutions should consider the tendencies of learners' self-directed learning and online learning attitudes when promoting and implementing online learning and teaching. Schools and teachers should cultivate learners' self-directed learning and online learning attitudes to enhance their learning behavior in online learning environments.

The results of this study indicate that self-directed learning is a key factor in the learning behavior of both Taiwanese and Chinese learners in online learning environments. In addition, attitudes towards online learning are essential elements of learning behavior. Therefore, learners must develop the ability for self-directed learning in order to effectively self-learn, and possess the attitudes necessary for online learning, such as confidence in using computers and smartphones to complete online courses.

Since the global pandemic of COVID-19 in 2020, learners have been unable to acquire knowledge and skills through physical classes. Schools and educational institutions have therefore established online teaching environments to allow students and learners to continue learning. China was one of the first countries to experience the outbreak of the pandemic, and therefore, schools and educational institutions at all levels in China have implemented online teaching and learning to assist learners in their online learning. This research targets the successful implementation of online teaching in China, effectively exploring the important factors and complex relationships involved in the learning behavior of online learners, and creates a theoretical framework for this study. It also provides scholars and experts in the field of education with a better understanding of the learning behavior of online learners, which can be used to develop friendly online learning strategies and environments.

## 5. Conclusions

Learners' self-directed learning and online learning attitudes are complex, multi-criteria indicators of competence that cannot be precisely defined and measured, and there are the criteria are characterized by complex and entangled relationships. The results of this study show that the dimension of self-directed learning influences the dimension of online learning attitudes. The criteria for self-directed learning and online learning attitudes are correlated with each other and the degree of influence on the online learning attitudes varies among the criteria.

In terms of the dimension, firstly, self-directed learning influences online learning attitudes. In terms of the dimension level, self-directed learning is the cause that influences dimensions, and online learning attitudes are the effect that is influenced by it. Therefore, to strengthen learning behavior in online learning, learners can start by constructing a self-directed learning dimension. Secondly, self-directed learning is the main determinant dimension of learners' learning behavior, it directly influences online learning attitudes, and is a fundamental factor in enhancing learners' learning behavior. Therefore, learners need to develop self-directed learning to establish the foundation of their learning behavior in online learning.

In the criteria section, firstly, self-learning efficacy, computer/smartphone and internet confidence, online learning interest, computer/smartphone usage, and computer/smartphone preference are the main influencing criteria for other criteria. In particular, the computer/smartphone preference is the most influential criterion, and the self-learning efficacy, computer/smartphone and internet confidence, online learning interest, computer/smartphone usage, and computer/smartphone preference, criteria affect each other and also affect other criteria. In addition, the computer/smartphone preference criterion is the strongest influencing criterion for other criteria. Learners can start with the strongest

and most influential computer/smartphone preference to enhance their online learning attitudes by encouraging learners to enjoy accessing and operating computers and smartphones. Learners can also enhance self-directed learning through self-learning efficacy to develop skills for continuous learning and efficient learning, as well as other skills.

Self-learning efficacy, computer/smartphone and internet confidence, online learning interest, computer/smartphone usage, and computer/smartphone preference are the main influences on the other criteria of online learning attitudes and self-directed learning. Therefore, learners should have the skills of computer/smartphone and internet confidence, online learning interest, computer/smartphone usage, computer/smartphone preference, etc. Furthermore, learners need to develop self-learning skills. Secondly, self-learning efficacy, computer/smartphone and internet confidence, online learning interest, computer/smartphone usage, and computer/smartphone preference are key determinants of online learning attitudes and self-directed learning. Learners should be able to grasp learning opportunities and overcome barriers to learning; learners should be confident in their learning abilities and computer performance, smartphones and the internet; learners should enjoy and look forward to learning online; learners should be able to use computers and smartphones in their studies, life and work and enjoy accessing and operating them.

The determinants and interactions of online learning attitudes and self-directed learning have not been explored extensively in previous studies; online learning attitudes and self-directed learning are important dimensions that influence learners' learning behavior. In addition, scholars who study online learning attitudes and self-directed learning have different theoretical perspectives. To understand the problems mentioned, this study combines the fuzzy Delphi method and the fuzzy DEMATEL method to propose a more comprehensive and complete set of determinants of self-directed learning and online learning attitudes. There is no research paper on this subject, so this study has academic value. In summary, the academic value of the findings of this study is as follows: 1. The study integrates theoretical perspectives on self-directed learning and online learning attitudes and uses a wide range of perspectives to collect and analyze the relevant literature to select indicators of self-directed learning and online learning attitudes and to identify the dimensions and criteria of self-directed learning and online learning attitudes by integrating the views of researchers and experts in online learning; 2. Using the fuzzy DEMATEL method to evaluate the dimensions and criteria of self-directed learning and online learning attitudes, the cause–effect diagrams were computed and analyzed to provide a clear and easy understanding of the complex cause–effect structure among the dimensions and criteria of self-directed learning, online learning attitudes and the strength and extent of the influence of these factors.

In terms of practical implications, the findings of this study reveal a number of important implications for the learning behavior of learners in online learning. Schools, educational institutions, and teachers can use the results of this study to identify the structural interrelationships and causal relationships among the indicators of learners' self-directed learning and online learning attitudes and to select the most important key indicators of self-directed learning and online learning attitudes, which will help schools, educational institutions, and teachers to understand learners' learning behavior in online learning programs, targeting learners' self-directed learning and online learning attitudes, and improving online learning programs. This will help schools, educational institutions, and teachers understand learners' learning behavior in online learning programs, focusing on the key criteria of learners' self-directed learning and online learning attitudes, improving online learning programs, cultivating the key criteria of learners' self-directed learning and online learning attitudes, which can effectively enhance learners' self-directed learning and online learning attitudes, and potentially improve learners' learning behavior and learning outcomes. In addition, this study provides a visualization chart to identify the relationships and determinants between learners' self-directed learning and online learning attitudes.

Because this study mainly aims to explore and analyze the effects that online learning has on learners' learning behavior through the viewpoints of scholars and experts applying online teaching at universities. Therefore, the research object is mainly scholars and experts; this study adopts experts' investigative approaches in exploring and analyzing. However, learners were not the object of research. Future research can address this limitation. Additionally, the research object of this study is scholars and experts from western Taiwan, Beijing and Tianjin in China. However, it is worth exploring and analyzing data from other geographical locations, which will vary due to factors regarding conditions in different countries, information technology, etc.

The results of this study could provide meaningful references for online education researchers, practitioners, and learners regarding their intrinsic learning abilities, intentions, and tendencies in online learning environments. As this study was limited to surveying scholars and practitioners from China and Taiwan, future research could involve a more diverse range of researchers, practitioners, or learners from different countries. This would ensure that online learning designs not only keep pace with technological developments but also fully consider important indicators of learners' intrinsic learning.

**Author Contributions:** Conceptualization, J.L.; Methodology, J.L.; Validation, C.-H.W.; Investigation, C.-H.W.; Resources, J.L.; Data curation, C.-H.W.; Writing—original draft, J.L.; Writing—review & editing, C.-H.W. All authors have read and agreed to the published version of the manuscript.

**Funding:** This research received no external funding.

**Institutional Review Board Statement:** Not applicable.

**Informed Consent Statement:** Not applicable.

**Data Availability Statement:** All data used for research are included in the content of the article and in Tables 1–5.

**Conflicts of Interest:** The authors declare no conflict of interest.

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
