# Peer review of "Determinants of Learners’ Self-Directed Learning and Online Learning Attitudes in Online Learning"

_sustainability, doi:10.3390/su15129381_

Round 1
Reviewer 1 Report
I congratulate the authors for the work done, while I offer some comments that I hope are understood from a constructive perspective:
On line 100 there is information that seems to be repeated.
Point 3 should be described more precisely, organizing the information into sections (participants, instruments, procedure, and analysis) with a detailed description in each one of them in such a way that the reader is offered the possibility of being able to replicate the findings.
It is spoken in different places of interest and the objective of the work (e.g. line 125 and following or line 140). However, a paragraph is missing where the main objective of the work is clearly formulated. It should be placed just before starting section 3, since the method must be aligned with the proposal of objectives.
Review wording line 143
The information of the procedure is mixed with the presentation of the results.
Both the procedure and the data analysis should be described with greater order and clarity.
The results should announce the tables in an orderly manner, describing the findings of each before presenting the next. On the other hand, the results must be described more clearly, accompanying the statements with the data.
There should be a section for discussion and another for conclusions, in such a way that the information in each section is clearly defined. In the first case, relating the findings with the previous literature and in the second, recapitulating the implications.
Finally, the bibliographical references should be updated.
Kind regards
Author Response
Dear Reviewer,
Thank you very much for your valuable comments. This manuscript has been revised according to your opinions, which makes this manuscript more perfect and complete.
- Duplicate words have been removed.
- In the third section research methodology and design, there are added research steps, calculation equations and analysis, etc.
- The main objective of the work is clearly formulated has been added in the abstract, and the method is aligned with the proposal of objectives has been added in the third section.
- Original wording of line 143 has been reviewed and corrected.
- Section 3 focuses on the research process and section 4 presents the results of the analysis.
- Supplementary descriptions of the procedure and the data analysis in a procedural and clearer manner in sections 3 and 4.
- In the fourth section, the research results are analyzed and explained in an orderly manner, and tables are attached, with quadrant diagrams and supporting data of the tables, and the research results are clearly described.
- In section 4, 4.1 and 4.2 were added to highlight the correlation and implications of our research results with past literature.
- References updated.

Reviewer 2 Report
The research area is interesting, however, there are certain flaws that need to be addressed. For instance, the title is too long and uninteresting, and somewhat grammatically incorrect. Each paragraph has a number of variable names, rather each sentence is 'over-crowded' with terms, which makes it dull and boring to read. Secondly, the study is not supported by recent literature which reduces its credibility. Moreover, it lacks theoretical underpinning. Thirdly, the research design and methodology are not clearly stated; and there is a sudden jump to results and conclusions without any comprehensive inferential analyses. Also, the language used needs extensive improvement, as the sentences are not structured well and there is too much repetitive content.
Author Response
Dear Reviewer,
Thank you very much for your valuable comments. This manuscript has been revised according to your opinions, which makes this manuscript more perfect and complete.
- The title has been corrected to be more concise; paragraphs have been rearranged to make them more interesting to read.
- Several recent literatures (including 2020 to 2023) have been supplemented to strengthen the credibility of this study, including 3 literatures in 2021, 10 literatures in 2022, and 1 literature in 2023.
- The design and methodology of this study are based on the fuzzy multi-criteria decision-making method based on the research topic, so the research steps, derivation and equations are supplemented in section 3 to support the reasoning analysis. In addition, each paragraph has been re-examined and corrected, and some duplicated content has been deleted.

Reviewer 3 Report
This study is somewhat innovative, but has some problems.
1. Many expressions in the manuscript do not conform to the specification, e.g. lines 99-100.
2. The authors have not taken into account the specifics of the study sample, which is crucial for the study.
3. The authors take the case of online education in Tianjin and Taiwan as the research object, whether there are differences between the two, these need to be explained in detail by the authors. In addition, this study should highlight the specificity of online education in China, otherwise the value of the study will be reduced.
Author Response
Dear Reviewer,
Thank you very much for your valuable comments. This manuscript has been revised according to your opinions, which makes this manuscript more perfect and complete.
- Original wording of lines 99-100 have been reviewed and corrected. In addition, each paragraph has been re-examined and corrected, and some duplicated content has been deleted.
- In section 2, add 2.2 Fuzzy Delphi method, 2.3 Fuzzy DEMATEL method to supplement the collection of research samples. In addition, in the third section, the research sample information of supplementary experts and scholars is added.
- In section 4.2, it is supplemented to explain whether there are differences in the learning behaviors of learners in China and Taiwan in the online education environment. In addition, it also adds that China's online education can provide an opportunity to construct an online learning environment and improve learners' learning behavior.

Reviewer 4 Report
The article deals wtih the determinants of learners' self-directed learning and on-line learning attitudes in on-line learning. The paper has potential but needs substantial changes.
The abstract should be more informative, with a clear statement of the aim of the paper, the motivation of the paper and a clear presentation of the findings.
The introduction suffers from a lack of transparent and comprehensible context for the discussion in the body of the paper. Motivation and research gap should be strengthened.
Theoretical part needs improvement. At the moment this part suffers from lack of explicit keynote. I suggest to rewrite this part to be comprehensive, complex and logical and to be more related to the title of the paper, the aim and the research. There is a noticeable lack of hypotheses and their development.
In the research methodology part, the Authors do not provide the reader with sufficient information. I suppose that if the Authors add hypotheses, it will be easier to present the research methodology properly.
Findings and discussion need rewriting. I encourage the Authors to rewrite the findings in a more complex, logical and comprehensive way in relation to the aim, title of the paper and added hypotheses.
The conclusion section needs to add limitations of the study.
Author Response
Dear Reviewer,
Thank you very much for your valuable comments. This manuscript has been revised according to your opinions, which makes this manuscript more perfect and complete.
- Based on the valuable suggestions provided by the reviewer, substantial changes were made from the abstract to the discussion.
- In the abstract section, there is a supplementary statement of the purpose of the paper, the motivation of the paper, and a clear statement of the finds.
- In section 1, the introduction discusses the main body of the paper in a transparent and understandable context and reinforces the motivation and research gaps of this study.
- in section 2, the construction of the theory has been strengthened and the theoretical cornerstone is clear, including the title and purpose of this research, which are more relevant to the title and purpose of this study. In Section 3, it is added that the use of multi-criteria decision analysis to replace the statistical inference of general hypotheses is more in line with comprehensive, complex, logical concepts.
- In the third section research methodology and design, there are added research steps, calculation equations, analysis, etc., and add why Fuzzy DEMATEL is used instead of general hypotheses for statistical inference results.
- In section 4, the findings and section 5 discussions are enhanced and supplemented to explain the research results and implications. The topic of this research paper is the decisive factor, so the multi-criteria decision analysis method is more suitable for exploring and analyzing complex multi-factor relationships and key factors. The multi-criteria decision analysis in this study uses the Fuzzy DEMATEL method and the Fuzzy Delphi method. Although the Pearson correlation coefficient test can explore and analyze the relationships between factors, it is not able to clarify what relationships are among factors or complex relationships among factors. Nevertheless, Fuzzy DEMATEL can conquer the problems. Therefore, in section 3 there is a supplementary explanation of why Fuzzy DEMATEL is used, without using general hypotheses and statistical inference results
- In the conclusion section, there is a supplementary explanation of the limitations of this study.

Round 2
Reviewer 2 Report
Well done.
Author Response
Dear Reviewer,
Thank you very much for accepting this manuscript. To further improve the manuscript, some adjustments have been made to the abstract, and additional explanations have been provided in sections 1, 4, and 5.

Reviewer 3 Report
The manuscript has been carefully revised, but should highlight the specificity of online education in China, compared to Western countries.
Author Response
Dear Reviewer,
Thank you very much for your valuable comments. This manuscript has been revised according to your opinions, which makes this manuscript more perfect and complete.
1. In the first section, there have been additional supplementary explanations regarding the comparison and differences between online education in China and Western countries. It also highlights the importance of self-directed learning and attitudes toward online learning in the online learning environment.
2. To further improve the manuscript, some adjustments have been made to the abstract, and additional explanations have been provided in sections 1, 4, and 5.

Reviewer 4 Report
Despite the efforts made by the Authors to improve the article, the changes made are not enough to improve the quality of the manuscript and do not change my recommendation.
Author Response
Dear Reviewer,
Thank you very much for your valuable comments. This manuscript has been revised according to your recommendations, which makes this manuscript more perfect and complete.
1. To further improve the manuscript, some adjustments have been made to the abstract, and additional explanations have been provided in sections 1, 4, and 5.
2. The topic of this study is exploratory research, rather than adopting a confirmatory research approach that requires setting hypotheses and validating them.

Round 3
Reviewer 4 Report
The changes made are not sufficient to improve the quality of the manuscript and do not change my recommendation.
Author Response
Dear Reviewer,
Thank you very much for your valuable comments. This manuscript has been revised, which makes this manuscript more perfect and complete.
1.It has been confirmed that the references are correct, and the original reference 110 is replaced into Cifuentes-Faura, J., Faura-Martínez, U., & Lafuente-Lechuga, M. (2020).
2.References in the manuscript have been reordered and numbered.
